# Latent Heat Storage and Thermal Efficacy of Carboxymethyl Cellulose Carbon Foams Containing Ag, Al, Carbon Nanotubes, and Graphene in a Phase Change Material

**DOI:** 10.3390/nano9020158

**Published:** 2019-01-28

**Authors:** Hong Gun Kim, Yong-Sun Kim, Lee Ku Kwac, Hee Jae Shin, Sang Ok Lee, U Sang Lee, Hye Kyoung Shin

**Affiliations:** Institute of Carbon Technology, Jeonju University, 303 Cheonjam-ro, Wansan-gu, Jeonju-si, Jeollabuk-do 55069, Korea; hgkim@jj.ac.kr (H.G.K.); wva223g6@naver.com (Y.-S.K.); kwack29@jj.ac.kr (L.K.K.); ostrichs@jj.ac.kr (H.J.S.); lso0594@naver.com (S.O.L.); 2dbtkd@naver.com (U.S.L.)

**Keywords:** carbon foam, nanomaterials, phase change material, thermal conductivity, latent heat storage

## Abstract

Carbon foam was prepared from carboxymethyl cellulose (CMC) and Ag, Al and carbon nanotubes (CNTs), and graphene was added to the foam individually, to investigate the enhancement effects on the thermal conductivity. In addition, we used the vacuum method to impregnate erythritol of the phase change material (PCM) into the carbon foam samples to maximize the latent heat and minimize the latent heat loss during thermal cycling. Carbon foams containing Ag (CF-Ag), Al (CF-Al), CNT (CF-CNT) and graphene (CF-G) showed higher thermal conductivity than the carbon foam without any nano thermal conducting materials (CF). From the variations in temperature with time, erythritol added to CF, CF-Ag, CF-Al, CF-CNT, and CF-G was observed to decrease the time required to reach the phase change temperature when compared with pure erythritol. Among them, erythritol added to CF-G had the fastest phase change temperature, and this was related to the fact that this material had the highest thermal conductivity of the carbon foams used in this study. According to differential scanning calorimetry (DSC) analyses, the materials in which erythritol was added (CF, CF-Ag, CF-Al, CF-CNT, and CF-G) showed lower latent heat values than pure erythritol, as a result of their supplementation with carbon foam. However, the latent heat loss of these supplemented materials was less than that of pure erythritol during thermal cycling tests because of capillary and surface tension forces.

## 1. Introduction

Thermal energy generated from fossil fuels is a type of high-quality energy that can be easily converted into mechanical energy and electric energy [1,2]. However, thermal energy of a low temperature, which remains after mechanical or electric use, is a low-level energy that is difficult to convert to other types of energy. For storage of high-quality thermal energy produced from low-level energy, the waste heat of low-level energy should be of a higher density to minimize energy loss [3,4,5]. Phase change materials (PCMs) capable of thermal energy storage (TES), especially latent heat storage, are increasingly receiving enormous amounts of attention as alternatives to physical thermal energy storage materials [6,7,8]. PCMs store energy in the form of latent heat during solid-liquid phase changes, and PCMs are widely used as thermal storage materials because they have a high thermal storage density, narrow range of temperatures for TES, and repeated utility [9,10,11,12]. Despite these advantages, PCMs typically have a low thermal conductivity, which can lead to disagreeable supercooling problems and phase segregation. To overcome this, high thermal conductive fillers have been widely used to enhance the thermal conductivity in PCMs, but high concentrations of fillers can reduce the TES capacity of the PCMs. Among fillers, porous thermal conducting materials are promising for resolving the above troubles related to the low thermal conductivity and reduction of TES; this can be accomplished through surface tension forces and capillary in the porous structure [13,14,15,16,17]. Especially, porous carbon materials can offer many advantages such as excellent physical, chemical, and thermal stability, non-toxicity, high specific surface areas, and fire resistance properties in addition to providing good heat storage capacities for PCMs [18,19]. Shin et al. [20] selected expanded graphites (EGs), which are one type of porous carbon material, as fillers and then researched the effects of the EGs on the thermal conductivity following impregnation of sodium acetate trihydrate (SAT) as the PCM. As shown by the results, the EGs increased the thermal conductivity of SAT composites and maximized the SAT impregnated between the interlayers of EGs. The resulting products exhibited excellent thermal cycling stabilities. Karthik et al. [21] prepared erythritol-graphite foam as a stable composite PCM and used the wetness impregnation method for the utilization and recovery of solar heat or industrial waste heat. Li et al. [22] researched the thermal performance enhancement of erythritol/carbon foam composites. By improving the wetting ability of the carbon foam surface by hydrogen peroxide, erythritol content was increased on the carbon foam surface, resulting in improvement of the thermal performances of the composites.

In this study, we prepared carbon foams as porous carbon materials to enhance the thermal conductivity and to maximize the latent heat of PCMs. Carbon foam has three-dimensional (3D) netlike construction and has been widely used in applications, such as electrode materials [23], heavy oil recovery [24], and sound energy absorption [25,26]. Here, carbon foam was prepared from carboxymethyl cellulose (CMC) derived from cellulose [27,28]. While CMC is an attractive material from eco-friendly and economic perspectives [29,30,31] to reduce the negative environmental effects [32], the thermal conductivity of CMC carbon foam is not very satisfactory. Therefore, to enhance the thermal conductivity of the CMC carbon foams used in the present study, we prepared various carbon foams containing various kinds of nano thermal conducting particles: Ag, Al, carbon nanotube (CNT), and graphene oxide (GO). We then investigated the enhancement effects on thermal conductivity. The reason for selecting these four nano thermal conducting particles was that carbon foams could not be prepared from CMC composites containing other nano thermal conducting particles due to decomposition by non-crosslinking between CMC and citric acid (CA) during electron beam irradiation (EBI). The erythritol as the PCM was added to the carbon foams containing the various nano thermal conducting particles via the vacuum method, and the different latent heat values and thermal properties were then evaluated. The carbon foams containing various nano thermal conducting particles and impregnated with erythritol were characterized by the use of differential scanning calorimetry (DSC) and scanning electron microscopy (SEM).

## 2. Experimental

### 2.1. Materials

The erythritol as the PCM was purchased from Cargill Co. (Minneapolis, MN, USA). The carboxymethyl cellulose used as the precursor of carbon foam was obtained from Sigma Aldrich Co. (St. Louis, MO, USA). The spherical Ag powder was received from HKK Solution Co. (Seoul, Korea); its degree of purity was 99.99%, with a particle size of 20 nm. The spherical Al powder was supplied from Ditto Technology Co. (Gunpo-si, Gyeonggi-do, Korea); its degree of purity was 99.99%, with a diameter in the range of 60–80 nm. The CNT powder was obtained from Nano Solution Co.; its degree of purity was above 95 wt %, with length in the range of 5–20 μm, diameter in the range of 8–15 nm, and tap density of 0.05 g cc^−1^. The GO powder with 65.33% carbon and 31.31% oxygen content was supplied from Graphenall Co. (Siheung-si, Guionggi-do, Korea); its D/G ratio was 1.04, with pH 2.87. The citric acid (CA) used as the cross-linking agent for preparing the carbon foam was of analytical grade and used as received.

### 2.2. Preparation of Simple CMC Carbon Foam and CMC Carbon Foams Containing Nano Thermal Conducting Materials

The simple CMC carbon foam and CMC carbon foams containing Ag, Al, CNT, and GO were respectively prepared through carbonization of CMC composites and CMC composites containing Ag, Al, CNT, and GO obtained via EBI treatments. For the simple CMC composite, 10 wt % CMC and 4 wt % CA were dissolved in distilled water at room temperature, which resulted in the production of a CMC paste. For CMC composites containing Ag, Al, CNT, and GO: 10 wt % CMC and 4 wt % CA were dissolved in distilled water at room temperature and then 2 wt % Ag, 2 wt % Al, 2 wt % CNT, and 2 wt % GO powder was added to the above pastes. The resulting pastes were stirred until the additives were uniformly dispersed. The obtained pastes were irradiated at a dose of 80 kGy of EBI, which was carried out by using a conveyor-type scanned beam with an accelerating voltage of 1.14 MeV and a beam current of 7.6 mA. The EBI treatment in this study was used for cross-linking of CMC and CA to prevent CMC composite decomposition at high temperature (over 1000 °C) during carbonization [27,28]. The EBI-treated samples were lyophilized. These lyophilized samples were carbonized at a high temperature of 1000 °C for 1 h under a nitrogen atmosphere in a tubular furnace to produce the carbon foams. These carbon foams were labeled as CF (CMC carbon foam), CF-Ag (CMC carbon foam containing 2 wt % Ag), CF-Al (CMC carbon foam containing 2 wt % Al), CNT (CMC carbon foam containing 2 wt % CNT), and CF-G (CMC carbon foam containing 2 wt % GO). In the case of the latter sample, GO powder in the CMC composite was reduced by thermal annealing during carbonization of CMC composite at 1000 °C, which changed it into reduced graphene oxide (rGO). Thermal reduction of GO is able to remove oxygen functionalities via a complicated mechanism that restores the π-conjugation arrangement characteristic of graphene. Thermal annealing at high temperature near 1000 °C has been proposed as the classic temperature for GO reduction resulting in rGO [33,34].

### 2.3. Impregnation of Erythritol into Carbon Foam and Carbon Foams Containing Various Nano Thermal Conducting Materials and Its Characterization

Figure 1 shows a schematic diagram of the process whereby erythritol was impregnated into the carbon foam, and carbon foams containing various nano thermal conducting materials, by the vacuum method to maximize thermal energy storage and minimize the loss of latent heat during repeated thermal cycling tests. The carbon foam and solid erythritol (1:100) were placed into an airtight glass container and this container was then heated in an oil bath. Molten erythritol at around 120 °C was impregnated into the carbon foam by removing the air, using a vacuum pump. The heating temperature was monitored by a thermocouple, which was located in a glass test tube filled with erythritol or erythritol mixed with carbon foams containing the various nano thermal conducting materials. The temperature variations of the erythritol in the glass test tubes during heating were passed through a temperature readout box, where the data were automatically recorded and saved onto a computer (Figure 1). The thermal conductivity measurements were conducted with a TPS 2500S instrument (Hot Disk, Göteborg, Sweden) and the resulting data obtained from the sensor, which was sandwiched between 2 same sample pieces. The samples were tested 10 times each, and the average value and the standard deviation of the obtained data were calculated. The latent heat values for pure erythritol and the various carbon foams impregnated with erythritol were determined by DSC (DSC25, TA instruments Co., New Castle, DE, USA) in the range of 30–140 °C. The samples were tested 10 times each, and the average value and the standard deviation of the obtained data were calculated. The microstructure for the morphology of various carbon foams and corresponding erythritol-impregnated carbon foams were evaluated with SEM (CX-200TA, COXEM, Daejeon, Korea).

## 3. Results and discussion

### 3.1. Thermal Conductivity of the Various Carbon Foams

Figure 2 shows the thermal conductivity of CF, CF-Ag, CF-Al, CF-CNT, and CF-G. As shown in Figure 2, the thermal conductivity values of carbon foam containing various nano thermal conducting materials were higher than that of the simple CF without any nano thermal conducting materials. Generally, the thermal conductivity of non-graphitized carbon foam will have a low value below 1 W mK^−1^ [35], and here, the thermal conductivity of CF without nano thermal conducting materials was 0.12 W mK^−1^. In contrast, the thermal conductivity values of CF-Ag, CF-Al, CF-CNT, and CF-G were approximately 0.18, 0.14, 0.22, and 0.27 W mK^−1^, respectively. Compared with CF, these values represented enhancements of approximately 150%, 116.67%, 183.33%, and 225%, respectively. Individually, Ag (429 W mK^−1^), Al (237 W mK^−1^), CNT (~3500 W mK^−1^), and graphene (4800~5300 W mK^−1^) are excellent thermal conducting materials. Therefore, the addition of Ag, Al, CNT, and graphene as nano thermal conducting materials into the carbon foam supported the development of substantially higher thermal conductivities.

### 3.2. Effects of Carbon Foams Containing Nano Thermal Conducting Materials on the Melting of Erythritol

Figure 3 shows the temperature changes with time during latent heat storage experiments with pure erythritol and erythritol added to simple CF, CF-Ag, CF-AL, CF-CNT, and CF-G. Here, we can see that erythritol initiated melting at a phase change temperature of around 120 °C and the time required to reach the phase change temperature of pure erythritol took around 1965 s. However, when erythritol was added to the simple CF and CFs with various nano thermal conducting materials, faster melting occurred than that with pure erythritol. In particular, as shown in Figure 3, when erythritol was added to the simple CF, the phase change occurred at approximately 1813 s. Even faster times of 1620 s, 1505 s, 1288 s, and 1046 s were achieved with CF-Ag, CF-Al, CF-CNT, and CF-G, respectively. These results indicated that the enhanced thermal conductivities of carbon foams led to reductions in the times required to reach the phase change temperature during heating with erythritol as the PCM, and among the carbon foams used in this study, CF-G showed the highest thermal conductivity and had the fastest phase change time, which was approximately 919 s less than that for pure erythritol.

### 3.3. Thermal Effects of Simple Carbon Foam and Carbon Foams Containing Various Nano Thermal Conducting Materials on the Latent Heat Storage of Erythritol

The latent heat for erythritol as the PCM was generally calculated by using the region of the endothermic peak near 120 °C during the DSC analysis. Figure 4a shows the DSC analysis results, and Figure 4b presents the latent heat values calculated from the DSC analysis. In Figure 4a, it can be seen that the intensity of the endothermic peaks when erythritol was added to simple CF, CF-Ag, CF-Al, CF-CNT, and CF-G were smaller than that of pure erythritol. This was because the amount of erythritol was reduced following the supplementation of the carbon foam. By comparing the latent heat values in Figure 4b, it can be seen that the latent heat value for pure erythritol was 358.04 ± 8.95 J g^−1^, but the latent heat values when erythritol was added to simple CF, CF-Ag, CF-Al, CF-CNT, and CF-G were generally similar, at around 290 and 300 J g^−1^. This was because of the similar impregnation amounts of erythritol by the vacuum method, resulting from the similar porous structure of carbon foams, regardless of the types of nano thermal conducting materials.

Figure 5 shows the results of the thermal cycling test (10 cycles) for pure erythritol and erythritol added to simple CF, CF-Ag, CF-Al, CF-CNT, and CF-G. These tests results indicated the thermal stability for the changes of latent heat during repeated melting and solidification cycles of the PCM. In the case of pure erythritol, latent heat values decreased continuously during 10 cycles and up to ~5% of the initial latent heat value was lost. However, when erythritol was added to simple CF, CF-Ag, CF-Al, CF-CNT, and CF-G, the materials exhibited much less latent heat loss than the pure erythritol. These results likely occurred because the porosity of carbon foam can prohibit the leakage of erythritol and thus minimize the latent heat loss during thermal cycling tests through capillary and surface tension forces. Therefore, it can be concluded that CF, CF-Ag, CF-Al, CF-CNT, and CF-G demonstrated great thermal reliability as fillers for PCMs.

### 3.4. Scanning Electron Microscopy (SEM) Images for the Impregnation of Erythritol into Carbon Foams

The impregnation of the PCM into a filler, such as carbon foam, has a very import role in maximizing latent heat storage. If the PCM is not impregnated into fillers properly, latent heat storage is reduced. Figure 6 displays SEM images of simple CF, CF-Ag, CF-Al, CF-CNT, and CF-G and the corresponding images for the carbon foams impregnated with erythritol. As shown in Figure 6, the morphologies of simple CF, CF-Ag, CF-Al, CF-CNT, and CF-G consisted of the microstructures of the respective carbon foams with various sizes and non-uniform porosities. Here, all of the images confirmed that respective carbon foams, after impregnation with molten erythritol by the vacuum method at around 120 °C, were successfully impregnated with erythritol without any voids. Therefore, it was clear that erythritol could be effectively impregnated into the pores of the respective carbon foams through the vacuum method.

## 4. Conclusions

In this study, we prepared both simple carbon foam without any nano thermal conducting materials and carbon foams containing Ag, Al, CNT, and graphene from CMC to enhance the thermal conductivity. We also used the vacuum method to impregnate erythritol as the PCM into the carbon foams to maximize the latent heat and to minimize the latent heat loss during thermal cycling. Firstly, we observed that CF-Ag (0.18 W mK^−1^), CF-Al (0.14 W mK^−1^), CF-CNT(0.22 W mK^−1^), and CF-G (0.27 W mK^−1^) carbon foams containing Ag, Al, CNT, and graphene, respectively, could support higher thermal conductivities than the simple CF (0.12 W mK^−1^) alone. Next, erythritol was added to the simple CF, CF-Ag, CF-Al, CF-CNT, and CF-G and this resulted in a decrease in the time required to reach the phase change temperature. Especially, when erythritol was added with CF-G, this material reached phase change temperature fastest and the time was around 919 s lower than that of pure erythritol. However, the addition of erythritol to the simple CF, CF-Ag, CF-Al, CF-CNT, and CF-G resulted in latent heat values between about 290 and 300 J g^−1^ lower than pure erythritol (358.04 ± 8.95 J g^−1^), which occurred as a result of the carbon foam supplementation. Regardless, the carbon foams displayed less latent heat loss than pure erythritol during thermal cycling tests. Lastly, with the SEM images, we could see that erythritol was effectively impregnated into the pores of the respective carbon foams through the vacuum method, and this helped to maximize the latent heat and minimize the latent heat loss during thermal cycling. Therefore, these carbon foams impregnated with erythritol represent promising materials for thermal energy storage applications. However, drawbacks in this study included: The difficulty in controlling the pore sizes uniformly of carbon foams; and the difficulty in preparing more varieties of carbon foams containing nano thermal conducting materials (because the CMC pastes containing some nano thermal conducting particles were not cross-linked between CMC and CA after treatment). Nevertheless, these carbon foams impregnated with erythritol represent promising materials for thermal energy storage applications.

## Figures and Tables

**Figure 1 nanomaterials-09-00158-f001:**
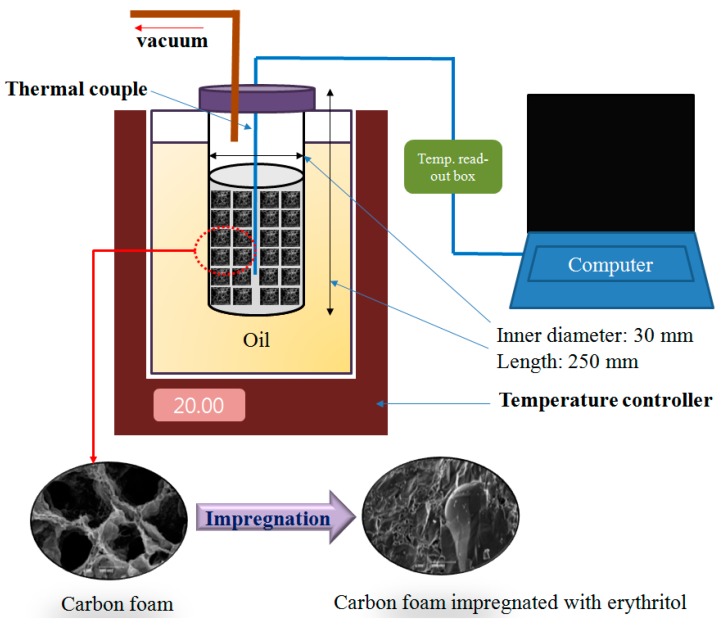
Schematic diagram of the temperature variation test unit for erythritol as the phase change material (PCM) and the vacuum impregnation of erythritol into carbon foam.

**Figure 2 nanomaterials-09-00158-f002:**
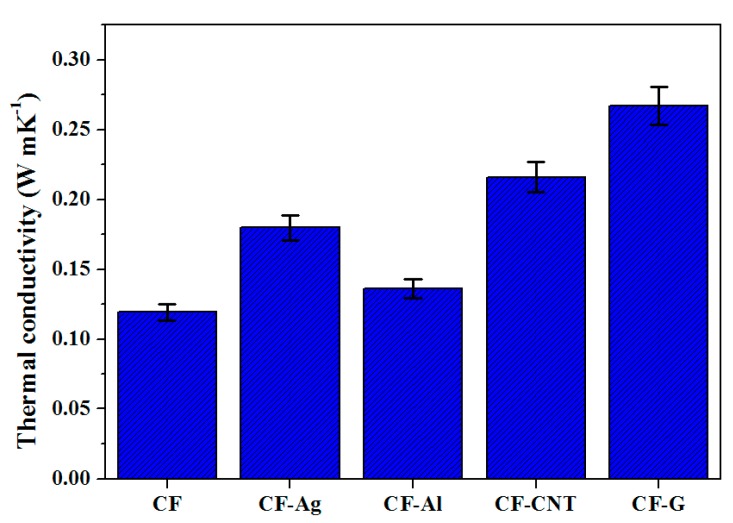
Thermal conductivity of simple carboxymethyl cellulose carbon foam (CF) and CF containing various nano thermal conducting materials.

**Figure 3 nanomaterials-09-00158-f003:**
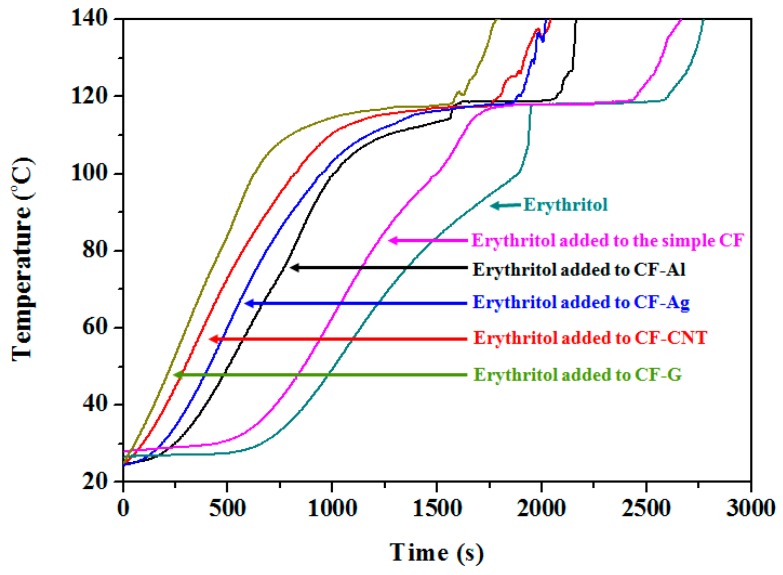
Temperature changes with time of erythritol and erythritol added to the simple CF and CF containing various nano thermal conducting materials.

**Figure 4 nanomaterials-09-00158-f004:**
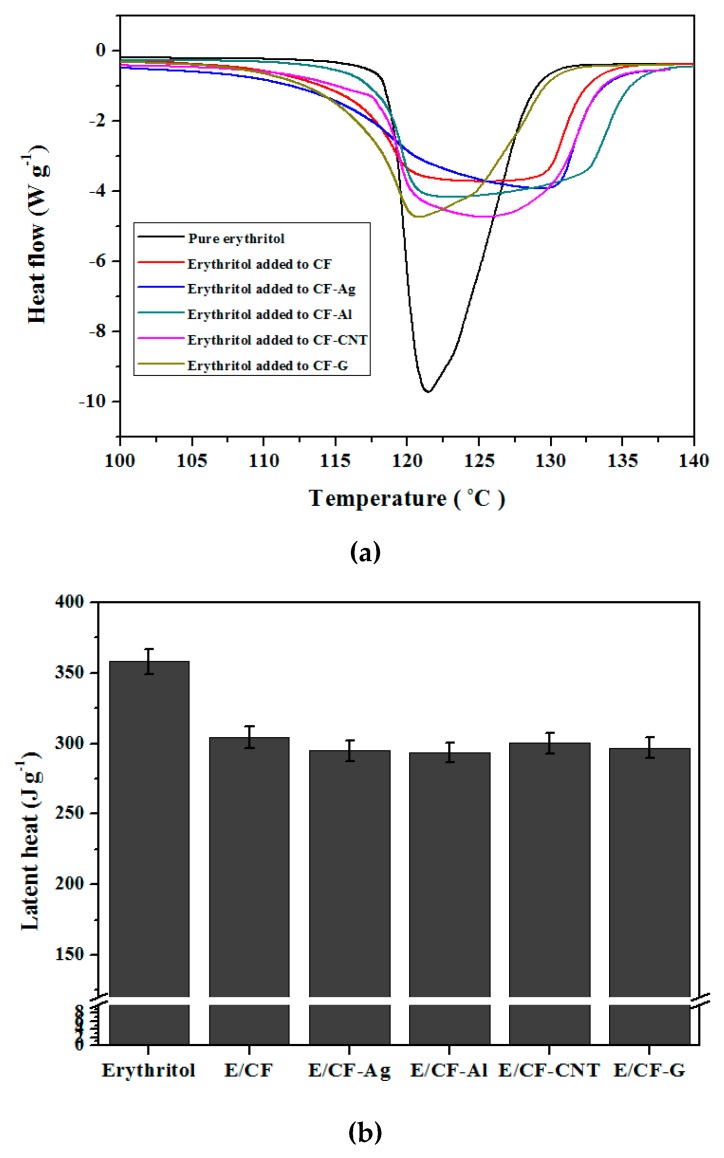
(**a**) Differential scanning calorimetry (DSC) analysis results and (**b**) comparison of latent heat values for pure erythritol and erythritol added to the simple CF and CF containing: Ag (CF-Ag); Al (CF-Al); CNT (CF-CNT); and graphene (CF-G).

**Figure 5 nanomaterials-09-00158-f005:**
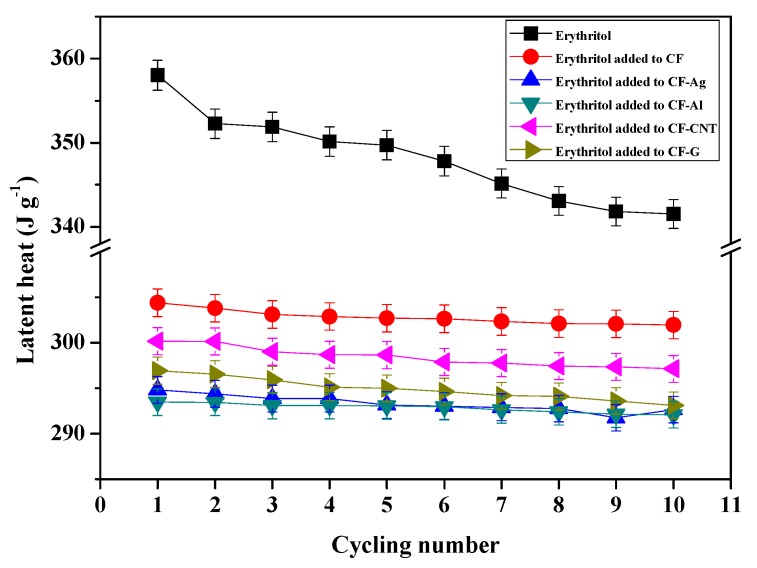
Thermal cycling test for pure erythritol and erythritol added to simple CF, CF-Ag, CF-Al, CF-CNT, and CF-G materials.

**Figure 6 nanomaterials-09-00158-f006:**
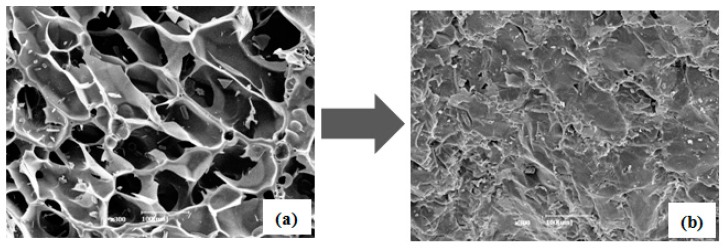
SEM images of (**a**) CF; (**c**) CF-Ag; (**e**) CF-Al; (**g**) CF-CNT; and (**i**) CF-G, as well as the SEM images for the corresponding carbon foams impregnated with erythritol (**b**,**d**,**f**,**h**,**j**, respectively).

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
