# Peer review of "Latent Heat Storage and Thermal Efficacy of Carboxymethyl Cellulose Carbon Foams Containing Ag, Al, Carbon Nanotubes, and Graphene in a Phase Change Material"

_nanomaterials, 2019, doi:10.3390/nano9020158_

Reviewer 1 Report

In this study, the authors have described how they prepared carboxymethyl cellulose carbon foams containing various fillers and impregnated them with erythritol as a phase change material. Although the results are interesting, the article has serious flaws, which must be fixed before reconsidering it for publication:

1) Experimental section is poorly described - firstly, it does not reveal how thermal conductivity was measured. Secondly, all these fillers should be characterized by SEM, Raman spectroscopy, particle size, etc. (where appropriate). For instance, you can have hundreds of types of CNTs depending on the number of walls, purity, catalyst content and so on. This has a significant effect on the end result, so should be disclosed.

2) To some readers it would be very informative to indicate why you irradiate the sample.

3) "Thermocouple" not "thermal couple".

4) Although some literature may report that treatment of GO in argon at high temperature is reduction, but this term should rather be used when hydrogen is the process gas. Please consider describing this as thermal annealing.

5) Please comment on the fact that these materials could in theory sediment in the CMC/CA paste, when dispersing agent or chemical functionalization are not introduced. Have you observed such effect?

6) How many samples were tested? According to Fig. 2 probably just one for each parameter because no standard deviation bars can be seen. If so, please repeat the study more times for each parameters to give insight about reproducibility of your method.

Author Response

Thank you for the reviewers for their very constructive comments on this manuscript.

Revision was done as follows.

1) Experimental section is poorly described - firstly, it does not reveal how thermal conductivity was measured. Secondly, all these fillers should be characterized by SEM, Raman spectroscopy, particle size, etc. (where appropriate). For instance, you can have hundreds of types of CNTs depending on the number of walls, purity, catalyst content and so on. This has a significant effect on the end result, so should be disclosed.

→ I revised as follows;

After:

For measuring the thermal conductivity: Thermal conductivity measurements were conducted with a TPS 2500S instrument, Hot Disk, Sweden and the resulting data obtained from the sensor which is sandwiched between 2 same sample pieces. Samples were tested 10 times each and average value and the standard deviation of the obtained data were calculated.

For fillers: As fillers in the carbon foams, spherical Ag powder was received from HKK Solution Co. and its degree of purity is 99.99%, with particle size of 20 nm. Spherical Al powder was supplied from Ditto Technology Co. and its degree of purity is 99.99%, with diameter in the range of 60-80 nm. CNT powder was obtained from Nano Solution Co. and its degree of purity is above 95 wt%, with length in the range of 5 – 20 μm, diameter ranging from 8 to 15 nm, and tap density of 0.05 g cc-1. GO powder with 65.33% carbon and 31.31% oxygen content was supplied from Graphenall Co. and its D/G ratio is 1.04, with pH 2.87 and XRD 2D value of 9.8.

2) To some readers it would be very informative to indicate why you irradiate the sample.

After:

EBI treatment in this study was used for cross-linking of CMC and CA to prevent CMC composite decomposition at high temperature (over 1000 °C) during carbonization [29-30].

3) "Thermocouple" not "thermal couple".

I revised into “Thermocouple”

4) Although some literature may report that treatment of GO in argon at high temperature is reduction, but this term should rather be used when hydrogen is the process gas. Please consider describing this as thermal annealing.

I revised as follows;

Before: GO powder in the CMC composite was reduced by using a high thermal treatment during carbonization of CMC composite at 1000°C, which changed it into reduce graphene oxide (rGO))

After: and CF-G (CMC carbon foam containing 2 wt% GO). In the case of the latter sample, GO powder in the CMC composite was reduced by thermal annealing during carbonization of CMC composite at 1000°C, which changed it into reduce graphene oxide (rGO). Thermal reduction of GO is able to remove oxygen functionalities via a complicated mechanism that restores the π-conjugation arrangement characteristic of graphene. Thermal annealing at high temperature near 1000 °C has been proposed as the classic temperature for GO reduction resulting in rGO.) [35-36]..

5) Please comment on the fact that these materials could in theory sediment in the CMC/CA paste, when dispersing agent or chemical functionalization are not introduced. Have you observed such effect?

CMC is used in various products, creams, lotions and toothpaste formulation where its good binding, thickening, and stabilizing properties are utilized. CMC is water-soluble materials and its viscosity is controlled according to CMC concentration which dissolved in distilled water. 10 wt% CMC paste in this study has high viscosity and then dispersed well without dispersing agent or chemical functionalization. In addition, nano particles as fillers are very light-weight and then were dispersed well in CMC paste.

6) How many samples were tested? According to Fig. 2 probably just one for each parameter because no standard deviation bars can be seen. If so, please repeat the study more times for each parameters to give insight about reproducibility of your method.

I revised as follows;

Samples were tested 10 times each and average value and the standard deviation of the obtained data were calculated. 

Figure 2. Thermal conductivity of simple CF and CF containing various nano thermal conducting materials.

Figure 4. (a) DSC analysis results and (b) comparison of latent heat values for pure erythritol and erythritol added to the simple CF, CF-Ag, CF-Al, CF-CNT, and CF-G materials.

Figure 5. Thermal cycling test of for pure erythritol and erythritol added to simple CF, CF-Ag, CF-Al, CF-CNT, and CF-G materials.

Reviewer 2 Report

The manuscript by Shin and co-workers details the preparation and application of tailored carbon foams. The topic is timely, has some sustainability aspect, and of interest to a broad audience. The work is sufficiently detailed to be reproducible. A good amount of data is presented in the manuscript to support the claims. However, there are several issues that must be addressed prior to publishing.

1. The title is long and therefore difficult to digest; a more succinct title would better assist the readers.

2. What was the rationale for the selection of the 4 materials to be incorporated into the carbon foams? Why Ag, Al, CNT and G were used and compared? Justification should be provided in the main text.

3. Avoid grouping excessing number of references, e.g. [1-8], [27-31]. Either discuss all articles individually, or mention the most important contributions.

4. Refrain from exaggeration and vague expressions, e.g. “enormous amounts”

5. What do the authors mean by “disagreeable super-cooling problems”? Rephrase as necessary.

6. The grade/purity of all chemicals and solvents should be provided under the materials section. The supplier for all materials should be given as well.

7. Line 87-88 needs to be rephrased.

8. Closely related recent research on erythritol/carbon foam for similar applications should be acknowledged (Carbon, 2015, 94, 266-276; IOP Materials Science and Engineering, 2017, 182, 012009).

9. Instead of the ambiguous x/y formatting of units, the IUPAC recommended x y^-1 format should be used throughout the manuscript.

10. The reproducibility of the Figures 2, 4–5 results should be demonstrated via including error bars on the plots. Their derivation, e.g. standard deviation, should be discussed in the text. How many repeats of the same materials were prepared and tested?

11. The caption for Figure 6 is somewhat confusing. Rephrase it in a more clear way.

12. The presented work relates to the emerging field of sustainable/green solutions. A short introductory paragraph should be added on the topic with some recent and broad examples to demonstrate the importance and extensive interest in finding sustainable/green solutions (ChemSusChem, 2018, 11, 3640-3648; Green Chem., 2018, 20, 4911-4919; Green Chem., 2017, 19, 3116-3125; Green Chem., 2018, 20, 4901-4910; ACS Catal., 2018, 8, 7430-7438).

13. The accuracy of the different measurement techniques employed in the study should be reported. In some cases, values are reported down to 0.01 decimal places, which seems inappropriate.

14. The conclusions section should summarize the main research findings, and preferably it should include some quantitative statements. The drawbacks and limitations of the methods should also be mentioned to add a critical edge.

Author Response

Thank you for the reviewers for their very constructive comments on this manuscript.

Revision was done as follows.

1. The title is long and therefore difficult to digest; a more succinct title would better assist the readers.

I revised into “Latent heat storage and thermal efficacy of carboxymethyl cellulose carbon foams containing Ag, Al, carbon nanotubes and graphene in a phase change material”

2. What was the rationale for the selection of the 4 materials to be incorporated into the carbon foams? Why Ag, Al, CNT and G were used and compared? Justification should be provided in the main text.

I revised as follows;

After: we prepared various carbon foams containing various kinds of nano thermal conducting particles: Ag, Al, carbon nanotube (CNT), and graphene oxide (GO); we then investigated the enhancement effects on thermal conductivity. The reason for selecting these four nano thermal conducting particles was that carbon foams could not be prepared from CMC composites containing other nano thermal conducting particles due to decomposition by non- crosslinking between CMC and CA during electron beam irradiation (EBI).

3. Avoid grouping excessing number of references, e.g. [1-8], [27-31]. Either discuss all articles individually, or mention the most important contributions.

As your comment I separated individually to avoid grouping number of references.

4. Refrain from exaggeration and vague expressions, e.g. “enormous amounts”

I deleted “enormous amounts” as your comment

5. What do the authors mean by “disagreeable super-cooling problems”? Rephrase as necessary.

I deleted “disagreeable” as your comment

6. The grade/purity of all chemicals and solvents should be provided under the materials section. The supplier for all materials should be given as well.

I revised as follows;

Before: Ag, Al, CNT, and GO powders used as fillers in the carbon foam were supplied from HKK Solution Co., Ditto Technology Co., Nano Solution Co., and Graphenall Co., respectively.

After: Spherical Ag powder was received from HKK Solution Co.; its degree of purity was 99.99%, with particle size of 20 nm. Spherical Al powder was supplied from Ditto Technology Co.; its degree of purity was 99.99%, with a diameter in the range of 60-80 nm. CNT powder was obtained from Nano Solution Co.; its degree of purity was above 95 wt%, with length in the range of 5 – 20 μm, diameter in the range of 8-15 nm, and tap density of 0.05 g cc-1. GO powder with 65.33% carbon and 31.31% oxygen content was supplied from Graphenall Co.; its D/G ratio was 1.04, with pH 2.87 and XRD 2D value of 9.8.

7. Line 87-88 needs to be rephrased.

I revised as follows;

Before: For CMC composites containing Ag, Al, CNT, and GO, 10 wt% CMC and 4 wt% CA were dissolved in distilled water at room temperature and then the appropriate materials were, namely, 2 wt% Ag, 2 wt% Al, 2 wt% CNT, 2 wt% graphene oxide, respectively; These samples were stirred until the additives were uniformly dispersed.

After:  For CMC composites containing Ag, Al, CNT, and GO, 10 wt% CMC and 4 wt% CA were dissolved in distilled water at room temperature and then 2 wt% Ag, 2 wt% Al, 2 wt% CNT, and 2 wt% GO powder was added to the above pastes; the resulting pastes were stirred until the additives were uniformly dispersed.

8. Closely related recent research on erythritol/carbon foam for similar applications should be acknowledged (Carbon, 2015, 94, 266-276; IOP Materials Science and Engineering, 2017, 182, 012009).

I revised as follow and added the reference [23] and [24]

After: Karthik et al. [23] prepared erythritol-graphite foam as a stable composite PCM and used the wetness impregnation method for the utilization and recovery of solar heat or industrial waste heat. Li et al. [24] researched the thermal performance enhancement of erythritol/carbon foam composites. By improving the wetting ability of the carbon foam surface by hydrogen peroxide, erythritol content was increased on the carbon foam surface, resulting in improvement of the thermal performances of the composites.

 9. Instead of the ambiguous x/y formatting of units, the IUPAC recommended x y^-1 format should be used throughout the manuscript.

I revised x/y used throughout the manuscript into x y^-1 as your comment

10. The reproducibility of the Figures 2, 4–5 results should be demonstrated via including error bars on the plots. Their derivation, e.g. standard deviation, should be discussed in the text. How many repeats of the same materials were prepared and tested?

I revised into the Figures including error bars and add as follows;

After: Thermal conductivity measurements were conducted with a TPS 2500S instrument, Hot Disk, Sweden and the resulting data obtained from the sensor which is sandwiched between 2 same sample pieces. Samples were tested 10 times each and average value and the standard deviation of the obtained data were calculated.

11. The caption for Figure 6 is somewhat confusing. Rephrase it in a more clear way.

I revised by adding the sentence;

The impregnation of PCM into filler such as carbon foam has very import role to maximizing latent heat storage. If PCM is not impregnated into fillers properly, latent heat storage is reduced.

12. The presented work relates to the emerging field of sustainable/green solutions. A short introductory paragraph should be added on the topic with some recent and broad examples to demonstrate the importance and extensive interest in finding sustainable/green solutions (ChemSusChem, 2018, 11, 3640-3648; Green Chem., 2018, 20, 4911-4919; Green Chem., 2017, 19, 3116-3125; Green Chem., 2018, 20, 4901-4910; ACS Catal., 2018, 8, 7430-7438).

I tried to revise by adding reference for the importance and extensive interest in finding sustainable/green solutions. However, it was very difficult to add an introductory paragraph relating to the emerging field of sustainable/green solutions in this manuscript. But, I did try my best to put a short but simple sentence

In line 68, I added “to reduce the negative environment effects” and added to reference No 34.

13. The accuracy of the different measurement techniques employed in the study should be reported. In some cases, values are reported down to 0.01 decimal places, which seems inappropriate.

I revised as your comment

After:

- pure erythritol was 358.04 ± 8.95 J g-1,

- these values represent enhancements of approximately 150%, 116.67%, 183.33%, and 225%, respectively.

14. The conclusions section should summarize the main research findings, and preferably it should include some quantitative statements. The drawbacks and limitations of the methods should also be mentioned to add a critical edge

I revised as adding quantitative statement and the drawbacks in the conclusion section as your comment

After:

- we observed that CF-Ag (0.18 W mK-1), CF-Al (0.14 W mK-1), CF-CNT(0.22 W mK-1), and CF-G (0.27 W mK-1) carbon foams containing Ag, Al, CNT, and graphene, respectively, could support higher thermal conductivities than simple CF (0.12 W mK-1) alone.

- However, drawbacks in this study included difficulty in controlling the pore sizes uniformly of carbon foams and in preparing more varieties of carbon foams containing nano thermal conducting (because the CMC pastes containing some nano thermal conducting particles were not cross-linked between CMC and CA after treatment). Nevertheless, these carbon foams impregnated with erythritol represent promising materials for thermal energy storage applications.

Round  2

Reviewer 1 Report

Thank you for including my suggestions. I recommend publication of the article in the present form.

Reviewer 2 Report

The authors have done a thorough revision, however the manuscript needs to be proofread as it has several typos and English errors.